# AgISM: A Novel Automated Tool for Monitoring Trends of Agricultural Waste Storage and Handling-Related Injuries and Fatalities Data in Real-Time

**Mahmoud M. Nour [1],\*, Yahia M. Aly [2] and William E. Field [1]**

[1]  Agricultural and Biological Engineering, Purdue University, West Lafayette, IN 47907, USA
[2]  Department of Management, Purdue University, West Lafayette, IN 47907, USA
\*    Correspondence: mnour@purdue.edu

**Abstract:** Availability of summarized occupational injury data is essential for establishing complete incident surveillance systems, targeting incident preventative efforts, assessing the efficacy of prevention programs, and enhancing workplace safety. There are currently limited automated injury monitoring systems for summarizing occupational injuries obtained from electronic news and other sources, or for visualizing real-time data through an output platform. A "near" real-time surveillance tool could enable researchers to visualize data as it is being collected and provide a more rapid monitoring method to identify patterns in injury data. An automated data pipeline method could provide more current, consistent, and reliable information for injury surveillance systems and injury prevention purposes. Such a system could help public policy makers, epidemiologists, and injury prevention professionals spend less time and effort on classifying cases, increase confidence in the data, and respond quicker to "patterns" of specific types of incidents. Currently, injury surveillance approaches generally rely on manual coding of injury data, resulting in inconsistencies in classification of incident, and contributing factors and considerable delays in publishing results. This study focused on developing and testing a more automated coding methodology for use with incident narratives for further data mining, analysis, and interpretation. The concept was tested on 491 documented fatalities or serious injuries involving agricultural waste storage, handling, and transport operations. The approach provided current and real-time summarization of incident data along with data analysis and visualization by using a standard questionnaire for record-keeping, Python data frames, and the MySQL database. Findings in this study provided evidence for the reliability of classifying injury news clipping narratives into external real-time incident categories. Results showed a very encouraging performance for the chosen model to monitor injury and fatality incidents with efficiency, simplicity, data quality, timeliness, and a consistent coding process.

**Keywords:** farm-related injuries; automated coding; injury case monitoring; fatality case monitoring; injury surveillance; prevention

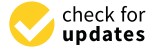



## 1. Introduction

Among all workplace environments, agriculture is one of the most dangerous occupations with alarming epidemiological evidence rates. High rates of mortality, morbidity, and near miss incidents have been identified including being crushed under overturned equipment, entanglement in machinery, drowning, suffocation, hearing loss, respiratory illnesses, and exposure to toxic levels of gases resulting from fermentation of forages and decomposition of livestock waste [1]. Currently, agricultural-related occupations do not have in place a comprehensive national centralized surveillance system for monitoring workplace injuries and fatalities. Work-related injury surveillance is essential for injury prevention efforts using an evidence-based approach. It is crucial for early identification of work injury clusters, mobilizing rapid response, and monitoring incident trends in agricultural workplaces. Such a system could assist by automating data acquisition and generation

of statistical alerts, monitoring injuries and fatalities in real-time or near real-time to detect hazards earlier than the traditional manual methods. Yet, gaps have historically existed and remain in national statistics, and support is needed to supplement current national surveillance to be more inclusive and comprehensive, recognizing the whole spectrum of agriculture-related injury and fatality cases [2]. According to Measure, (2014) much of the current information about work-related injuries, fatalities, and illnesses in the U.S. are recorded as short written narratives on electronic Occupational Safety and Health Administration (OSHA) logs, news clippings, published articles, and Workers' Compensation records [3]. Documenting, consistently coding, and summarization of these data are important for injury surveillance and prevention purposes but remain problematic. Currently, the process of assigning any coding methodology to the disarray of incident reports is often completed manually, extremely time consuming, prone to human error, and also suffers from quality issues such as inconsistencies in codes assigned by different coders [3]. On the other hand, recent automated models for data collection and record-keeping for monitoring cases have been shown to be highly effective, with good intuitive performance in many applications [3]. Current efforts to automatically classify workplace injury and fatality incidents have focused almost entirely on digital techniques.

Additionally, there does not appear to be an existing technique for efficiently capturing data about workplace injuries and fatalities from online news reports. Therefore, most occupational safety and health epidemiologists and safety professionals are continuing to use manual classification techniques. This involves searching the media, now almost exclusively online, or original reporting sources, reading the incident narrative report, coding the desired data and entering the data into some type of database, usually a spreadsheet. Introducing an efficient, more automated method for handling workplace injury data is likely to improve performance over traditional techniques that relied on numerous individuals manually monitoring injury and fatality cases, often for different purposes. Such a strategy could help improve efficiency, speed, and overall accuracy compared to traditional techniques [4]. The main functions of the proposed data collection and record-keeping system are to provide users with real-time data query and statistics, reduce the difficulty of manual coding systems, and to make the process run in a more stable manner and be lower in cost [5]. However, while it appears that manual coding of incidents can be simplified, or, at least, reduced in part, it is not feasible to completely eliminate this essential step from the process [6]. There will remain the need for an 'expert' to interpret the incident narratives to identify contributing factors.

In this study, incidents involving agricultural waste storage and handling facilities, transport equipment, and other waste-related operations, such as digesters and bio-gas generators have been documented as part of Purdue's Agricultural Confined Spaces-related Incident Database (PACSID). In this study, four primary objectives were identified in order to develop the semi-automated AgISM model for accurately storing and documenting injury cases within agricultural settings:

1.  Assess potential issues with conventional manual entry of selected data that are widely used and to enhance organization for easy retrieval of work-related incidents.
2.  Develop and test an automated pipeline using modern semi-automated techniques for monitoring trends of work-related incidents.
3.  Create a more robust method for obtaining longitudinal summaries of the selected data.
4.  Develop a dashboard visualization process for near-real-time workplace incidence data and trending categories for future prevention measures.

The remainder of the article is organized by first presenting the results from related studies in the Related Works section. The Materials and Methods section then provides details for the proposed AgISM model and how the 491 cases were monitored using the AgISM model. The results are presented in the Discussion and Evaluation section. Finally, the conclusions and future research needs are provided in the Conclusions section.

## 2. Related Works

Although limited studies have focused on the development of an efficient automated monitoring system capable of collecting and recording incidents and cases corresponding to injuries in workplaces, many researchers have expressed the need to move away from the traditional manual-coding approaches, conducted by numerous coders with a diversity of backgrounds, due to their subjective and inefficient results.

Recently, Patel et al. (2017) identified three major gaps for the documentation of non-fatal agricultural injuries in the U.S. as follows: (1) insufficient data quality attributed to non-response, measurement errors, and underreporting; (2) untimeliness of data processing, and (3) lack of flexibility to integrate with other existing systems [7].

Poor quality of traditional manual-coding approaches can lead to unreliable insights from injury data and can hamper injury surveillance and prevention efforts [8]. Currently, traditional monitoring systems that are widely used are inefficient as they are only capable of collecting and recording data with inaccuracies in the range of 72–88% [8].

Throughout the last decade, studies of agricultural work injury incident classification have depended entirely upon manual-coding approaches for collecting and recording datasets. In addition, researchers have utilized different coding systems such as, Farm and Agricultural Injury Classification (FAIC) and Occupational Injury and Illness Classification System (OIICS) which were not interchangeable [9]. As manual coding is subjective, there can be substantial variation in the codes assigned by human coders depending on their experience level, background, and level of comprehensiveness of the incident narrative or news clippings [10,11]. Additionally, there exist some types of injury codes that cover many categories, with some being very closely related which adds an additional layer of complexity for human coders to select the most appropriate category [10,11].

Gorucu et al. (2020) mentioned five essential themes regarding accurate and consistent coding of agricultural injury data obtained from AgInjuryNews.org reports: (1) inclusion/exclusion based on industry classification system; (2) inconsistent/cluttered reports; (3) incomplete/nonspecific reports; (4) effects of supplemental information on coding, and (5) differing interpretations of code selection rules (primary/secondary injury sources) [12]. These themes were considered and validated when the automated monitoring system was developed.

However, most agricultural injury data are not currently collected and recorded for analysis utilizing sources such as the Occupational Safety and Health Administration (OSHA), Census of Fatal Occupational Injuries (CFOI), the National Safety Council (NSC), the National Institute for Occupational Safety and Health (NIOSH), which initiated the National Traumatic Occupational Fatalities (NTOF) surveillance system, or the Farm and Agricultural Injury Classification (FAIC). A considerable amount of reliable and published data of specific types of agricultural fatalities have come from other sources and has been used for developing evidence-based safety prevention strategies. This includes: the BLS Current Population Survey (CPS), National Electronic Injury Surveillance System (NEISS) data, Purdue Agricultural Confined Space Incident Database (PACSID), and AgInjuryNews.org (AIN). A few states publish annual summaries of farm-related fatalities, such as Indiana, which has produced summaries for over 50 years [13].

Previous incident reporting and coding studies have almost completely lacked a systematization of a standardized automated monitoring method due to the lack of applicable computational tools. For this reason, there was a need identified for introducing automated surveillance of occupational-related incidents that is able to use near real-time workplace injury data observations. Lastly, utilizing an automated model for monitoring injuries and fatalities should help in more rapidly identifying and categorizing workplace injury trends.

## 3. Materials and Methods

The research conducted developed and tested a more reliable, computer-based model, designed to increase the accuracy and consistency of the data collection and summarization process for agricultural workplace injuries and fatalities. In this study, data from

491 agricultural waste-related injuries and fatalities were mined from the (PACSID). A dataset was acquired to test the proposed system and to help meet the three primary objectives that were identified. The proposed system was named "Ag Injury Surveillance and Monitoring (AgISM) Model". The overall pipeline is illustrated in Figure 1.

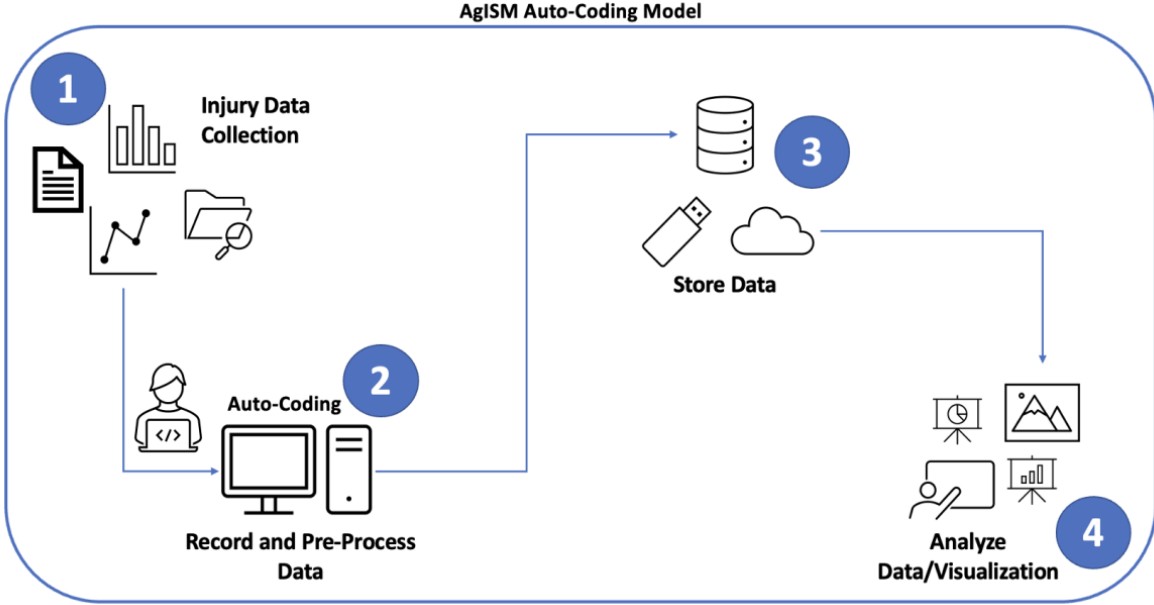

**Figure 1.** Overview of the validation process of AgISM model.

This database has been used to compile fatality and injury data related to agricultural confined spaces for over 45 years. Analysis of the selected data was tested by developing a novel pipeline model, capable of efficiently monitoring cases in an automated manner. The cases in the (PACSID) were collected and recorded manually between 1975 and 2021 with recent summaries published by Nour, et al., (2019, 2020, 2021) [14–18]. In the pre-processing stage of data mining, there were two concepts related to gathering and documenting relevant cases that were explored. This first is a Boolean logic model to select the specific cyber search terms used, and to identify key search factors and their relative significance. A Boolean logic model is a common tool on web search engines and refers to the logical relationship among multiple search terms. It provides the most effective principles of online searching and finds precision search results. The second was to test the applicability of the pilot coding scheme for choosing relevant fields for the coding process.

The automated record-keeping approach of managing electronic news clipping data was designed to maximize efficiency for inputting data, storing data, and summarizing and visualizing data. The basic structure and a schematic figure of the overall research methodology utilized is presented in Figure 1.

Five characteristics were considered to ensure data quality across each aspect of the pipeline: suitability, applicability, usability, reliability, and change management.

- Suitability—Is it appropriate for the amount and type of data being gathered?
- Applicability—As the needs of the project grow, is the designed system able to meet the new demands without significant modification?
- Usability—Does it require specialized training, or necessary insight or is it intuitive to operate?
- Reliability—Is the data cleaned and free of potential errors that could result in faulty conclusions?
- Change Management—How difficult is it to migrate from existing systems?

Each of these questions was addressed to ensure that the final product would meet the expectation of data users and is capable of meeting future needs. It also ensured that

ongoing insight, or interpretation of the data could be easily obtained with strong confidence in the data's accuracy. Furthermore, the design process considered what software and processes are currently in use, allowing for an efficient and reliable transition to an improved system [19].

For example, Microsoft Access as a database works for small projects with a limited number of users. It gives the ability to create forms and output charts without utilizing different software. It even gives the option to query the data for analysis. Hence, the data pipeline can be accomplished using one software package. However, if the project requires multiple users to access the data at the same time, large amounts of data, or advanced analysis, Microsoft Access presents serious limitations. It is important, therefore, to consider future requirements when picking software as the data scales, and the project needs to outgrow present problems.

## 4. Framework of the AgISM Model

In an effort to develop an unbiased, efficient, and robust real-time web tool for storing and recording injury cases, a semi-automated monitoring system was developed. The method's robustness was tested by combining multiple data sources and ensuring that the cumulative statistics align. Furthermore, there are various points of data validation. The first is the data entry ensuring the correct type and format of the input and that all the required fields are filled. The integration layer checks that the data are not duplicated, and that the data can be formatted according to the existing schema. This prevents issues when changing data sources. Finally, data inserted goes through anomaly analysis to highlight outliers, which could be the result of invalid entry or substantial variation of the source. Due to the schema design of clustering into predefined groups, it allows for the transformation of various narrative sources into quantitative categories. This reduces the uncertainty by highlighting missing data from each source, pointing outliers, and utilizing charts to demonstrate uncertainty. This is partly conducted by using a Boolean logic model. For example, a narrative of injury at a specific location involving family members or non-employees can be encoded to explain trends and correlations between incidents. This is implemented using visualization to cluster groups.

### 4.1. Dataset

For the purpose of this study, a dataset comprised of 491 different incidents/cases related to the storage, handling, and transport of agricultural waste that were previously documented were used in order to populate and test the AgISM model [18]. Since the 1970s, over 3000 cases involving agricultural confined spaces, including 491 incidents involving agricultural waste storage, handling, and transport, have been documented as part of ongoing surveillance by Purdue University's Agricultural Safety and Health Program (PUASHP). A Boolean logic model was used for selecting the specific cyber terms that were used as key search terms in order to obtain the data. There have been several efforts to examine this data; however, few have attempted to monitor or summarize, over time, the injuries and fatalities associated with agricultural waste storage, handling, or transport equipment and facilities [16].

### 4.2. Assess Potential Issues with Manual Entry of Selected Data and Enhance Organization for Easy Retrieval

No published work was identified that attempted to design or implement an agricultural-based surveillance method or consistent data classification/coding system that could be used to analyze cases involving agricultural waste-related injuries and fatalities. A uniform coding process was developed and applied to 491 individual cases. Ongoing surveillance of related injuries or fatalities was conducted throughout the study period. An estimation of the frequency and severity of these incidents, identifying geographic distribution, primary farm type, victim characteristics, and causative factors including those related to both respiratory and machinery hazards associated with agricultural waste

storage, handling, or transport were published [16]. The hazards associated with exposure to agricultural waste operations on farms are also included: drowning, exposure to pathogens, and fires and explosions due to production of methane gas.

### 4.3. Develop and Test a Novel Automated Tool for Monitoring Trends of Work-Related Incidents

An automated procedure was designed for collecting and recording data in a digital manner for monitoring injury and fatality cases. In order to input data, a Qualtrics "https://www.qualtrics.com (accessed on 1 February 2022)" survey was used to digitize the datasets. In this study, the Python programming language was used for parsing data from each survey, and uploading it onto the database for visualization. Utilizing a Qualtrics survey was found to be simpler to develop and maintain than a custom user interface as it reduces time and efforts. Additionally, in order to ensure data quality reliability and data validation, the proposed digital solution provided the ability to automate extracting data corresponding to different cases, and monitor the input data before inserting it directly into the database. This added an extra layer to ensure data integrity.

In order to develop the Qualtrics survey, different types of data collection techniques were utilized, namely, by providing choices to the users for standard data entries and by providing text boxes to further explain different complex cases. An example for the Qualtrics survey data input form is shown in Figure 2. Classification fields were created and were divided into multiple choices with an option to add "Other" in case none of the given options were applicable. This allowed the system to prevent typos or naming conventions that may prevent catching trends in the data.

**Figure 2.** Data input coding form of Qualtrics survey.

By providing users with options for data entry, the system is able to standardize the overall procedure. This is important as data consistency ensures that useful insight can be drawn when entering data. If multiple different terminologies are used by different users inputting data, then important conclusions will be inconsistent or missed on dashboards.

For example, some users inputting data might leave an unknown entry blank, and another might write unknown. Additionally, issues such as capitalization, and different naming conventions for categories can result in trends remaining undiscovered due to variations in reporting. Using a digital form prevents many of these issues becoming separate categories.

The input coding survey developed for providing users with a tool for inputting data consisted of a total of 19 questions that were asked in order to address the classification of key, previously identified, causative factors associated with agricultural waste storage, handling, and transport-related injuries and fatalities. A panel of experts were used to develop and prioritize these questions. The questions allowed either text input or by providing a set of multiple choices that could be chosen from. The following information was requested for each individual case:

1. First Name (text input)
2. Last Name (text input)
3. Age (text input)
4. Gender (multiple choice)
5. Cause of Incident (multiple choice)
6. Relationship to Farm (multiple choice)
7. Activities at the Time of Injury (multiple choice)
8. PACSID (text input)
9. Source of Data for Case Incident (multiple choice)
10. Data (text input)
11. Time of Case (text input)
12. Weekday of Incident (multiple choice)
13. Location of Incident (text input)
14. Total Number of Victims/Cases (text input)
15. Case Classification (multiple choice)
16. Type of Farm/Ranch (multiple choice)
17. Agent/Facility/Equipment / Involved (multiple choice)
18. Contributing Toxic Gases Identified (multiple choice)
19. Location of Case (multiple choice)

Example of screenshots from the different questions from the form are shown in Figures 3–5.

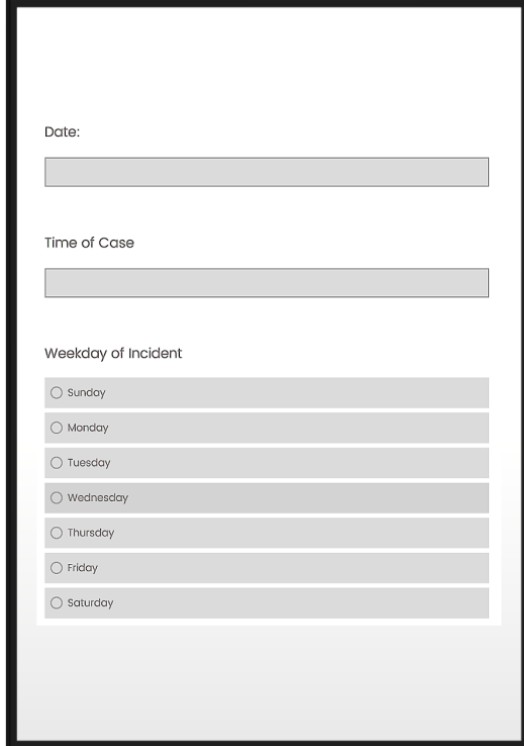
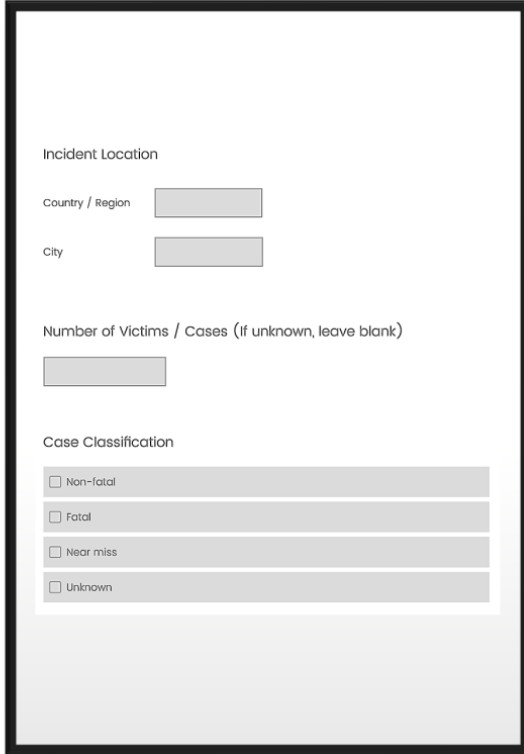

**Figure 3.** Qualtrics Survey Form: Date, Time, Day, Location, Victims, Class.

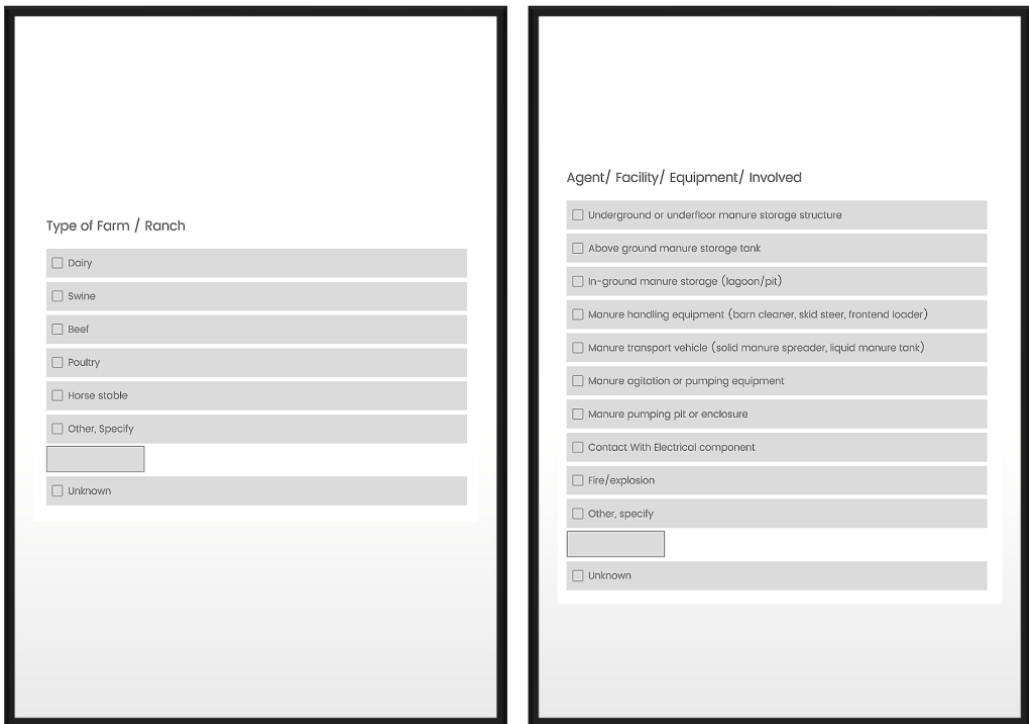

**Figure 4.** Qualtrics Survey Form: Farm Type and Agent involved.

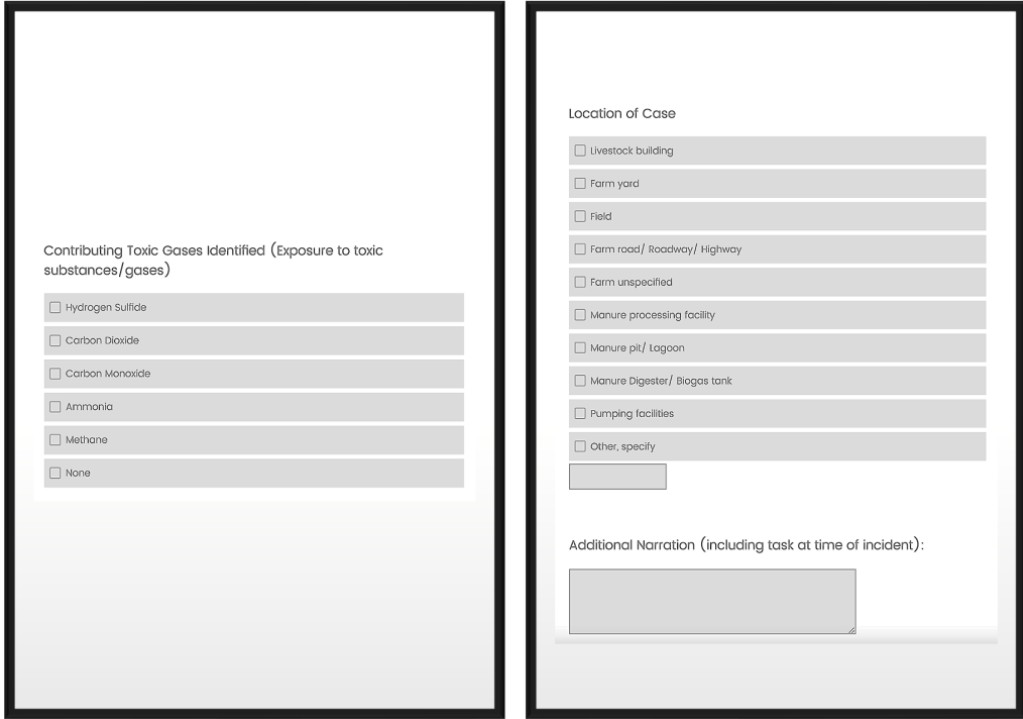

**Figure 5.** Qualtrics Survey Form: Toxic Gases, Location of Case, Additional info.

*4.4. Developing a More Reliable Simplistic Method for Obtaining Long-Term Effects of the Selected Data*

After using the Qualtrics survey to digitize data entry, the next step was to extract the data from the individual forms and manage the data within a database. Therefore, the Python programming language was utilized to clean and transfer data prior to organizing it within a central database. Furthermore, the tools provided by the programming language

were also used for data normalization. Although any modern programming language could be utilized to accomplish the task, Python is a robust high-level programming language that provides tools for easily managing data and for visualization. The Pandas Data frame package that is available for Python was used to parse data from Excel sheets to the database. In a case where there are large datasets consisting of hundreds of thousands or millions of data entries, C++ or Java might be a better choice. This is because C++ and Java are low-level languages which help run programs with greater efficiency through the use of algorithms and data structures. Nevertheless, Batch Normalization is the process of organizing data to minimize redundancy, and it allows relations to be created between data entities [20]. Demba laid out an algorithmic approach to normalizing the data by first removing redundant attributes through computing dependencies and then removing implied extraneous attributes. This requires prior domain knowledge to determine relationships between entities and hence determine keys to create the new tables. These relationships can be one-to-one (location), one-to-many (references), or many-to-many [20].

For example, an agricultural-related incident might involve five victims (as two incidents in the database did), and there are six different references or reporting sources where this information was obtained. Before data normalization, these would need to have as many rows as there are victims, and each row would have the repeated information of the case (location, details, etc.). Additionally, there would need to be at least six columns for references. In the future, if more information about the case is discovered through a seventh source, the information would need to change in five places and a seventh place would need to be added. This leads to extreme inefficiencies. Figure 6 provides an example of how part of the entire dataset was split into multiple tables for data normalization.

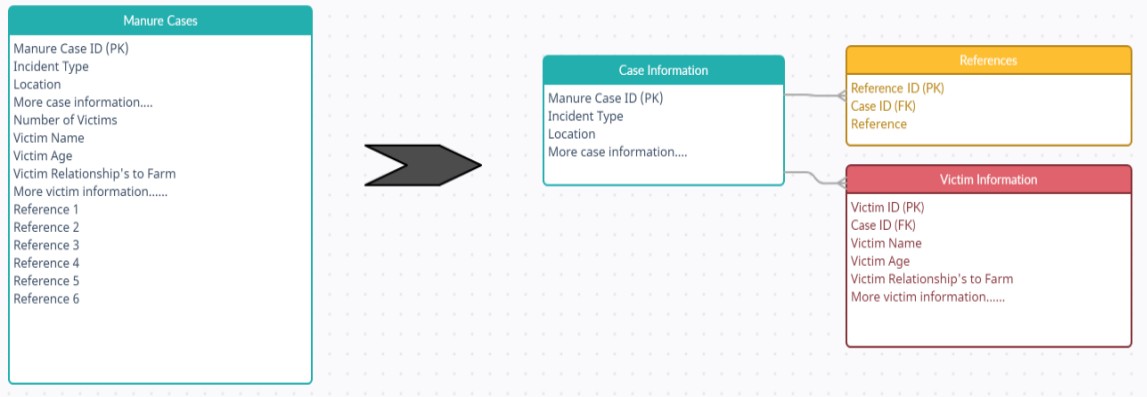

**Figure 6.** Subset of data normalization.

Multiple related tables are created from one large table which is responsible for the data entry. In this case, each case can have multiple references and multiple victims. If a change or a deletion is made, the change will only have to be made in one place and it will be reflected throughout. Once the data are cleaned and normalized, the dataset is migrated from Excel sheets to a database. The database that was chosen was a MySQL database. There are many options available, but as mentioned, the dataset is not large and (hence), a distributed database such as Hadoop or similar was not needed. Furthermore, MySQL integrates well with the chosen visualization tool (Tableau).

Storing data in a centralized database as opposed to individual Excel sheets, which is common in many domains, provided several advantages. First, databases provided the ability to scale as the dataset becomes larger. It also provided the capacity to maintain data integrity. For example, attempting to input text into a numeric column in a table will correctly return an error message. This ensured that errors passing through the data entry phase could be easily detected.

In this case in our study, an empty database was created, and Python was used to transfer the tables. Once this step was completed, the relationships were created between

the tables (cases and their references). From there, once new data were entered in Qualtrics, we simply appended to the correct tables after the data were checked for accuracy.

*4.5. Develop a Dashboard Visualization Process for Near Real-Time Workplace Incidence Data and Trending Categories for Future Prevention Measures*

Finally, a dashboard was developed to help visualize the data corresponding to different incidents. Multiple charts were created and hosted in an online dashboard using Tableau. Like other steps, visualization can be conducted in a wide array of software. Tableau was chosen due to the ability to easily connect to the database, create charts, combine charts into a dashboard, and host onto a server. Since the relationship between the tables is conducted in the databases, once a connection is established, plots were prepared that included fields from different tables. For example, a connection was made from location to number of cases so that the number of victims in each state could be determined rather than just number of cases.

The result is a live dashboard that displays the latest data in the database. Hence, as soon as new data are entered, new current charts are automatically updated. This is useful as a monitoring tool and triggers can be created to send notifications once the dashboard is updated. Figures 7–12 provide a sample dashboard that displays geographic location, age group of victims, fatality and non-fatality cases, and activities during the incidents of the 491 cases used in the study.

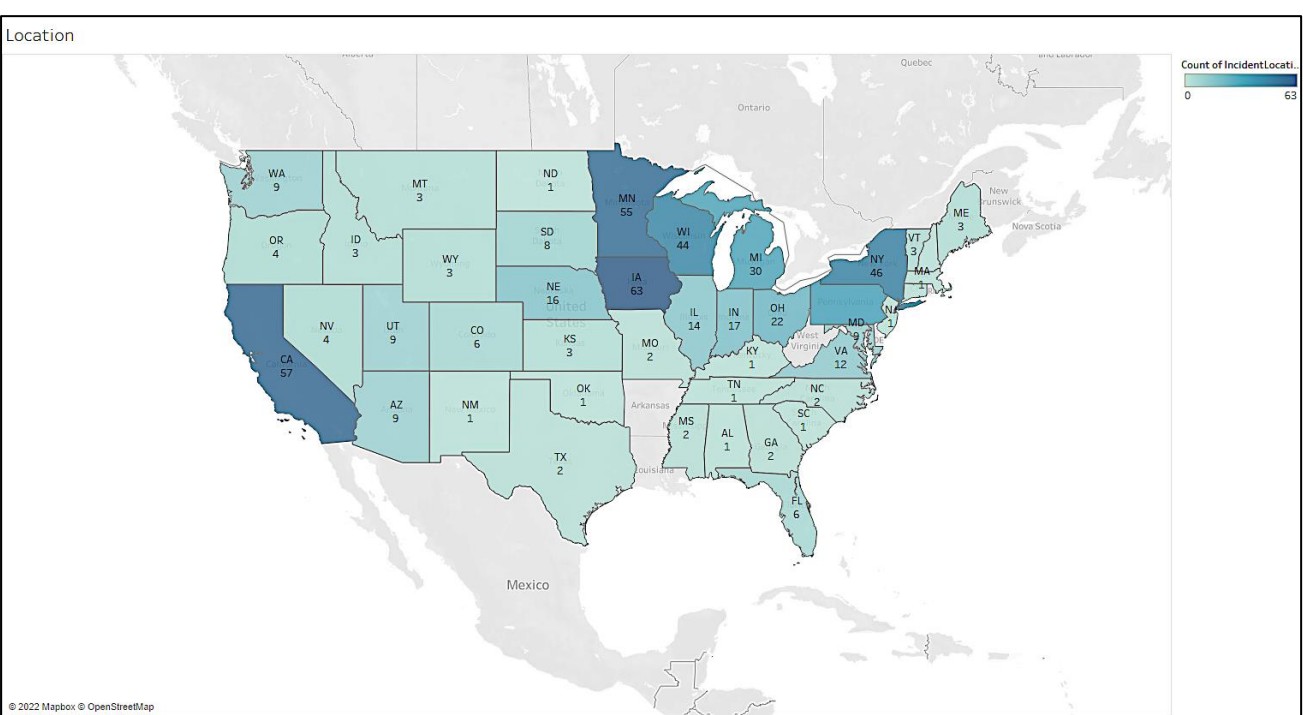

**Figure 7.** Dashboard software display map with case count.

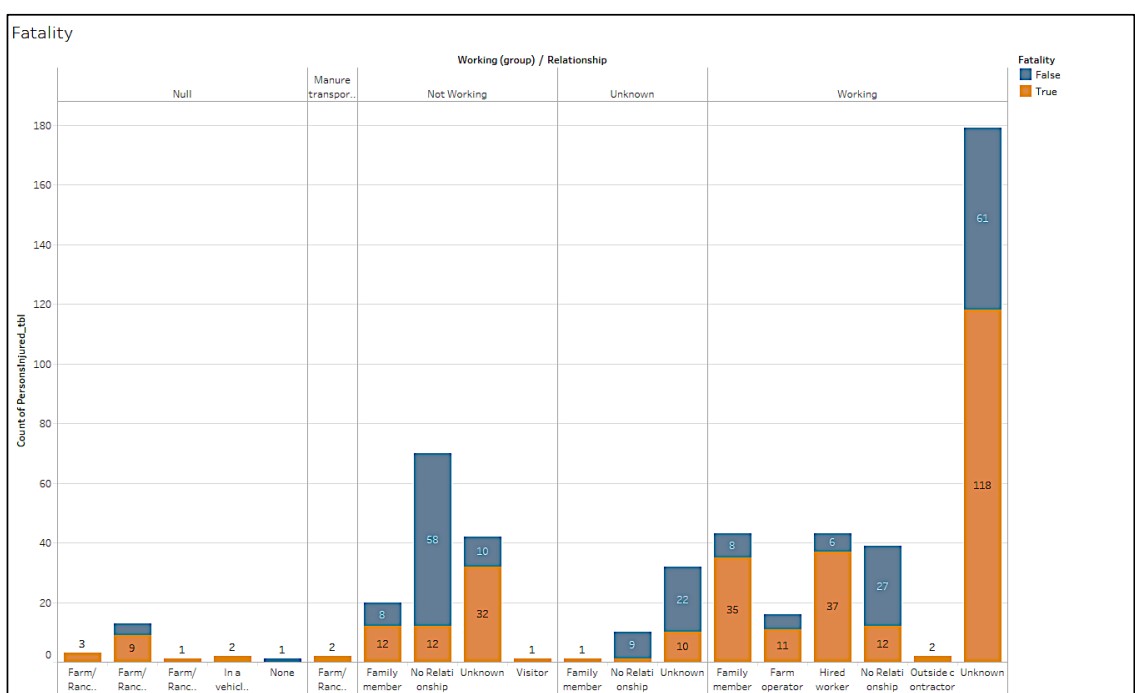

**Figure 8.** Dashboard software displays agricultural waste injury and fatality data.

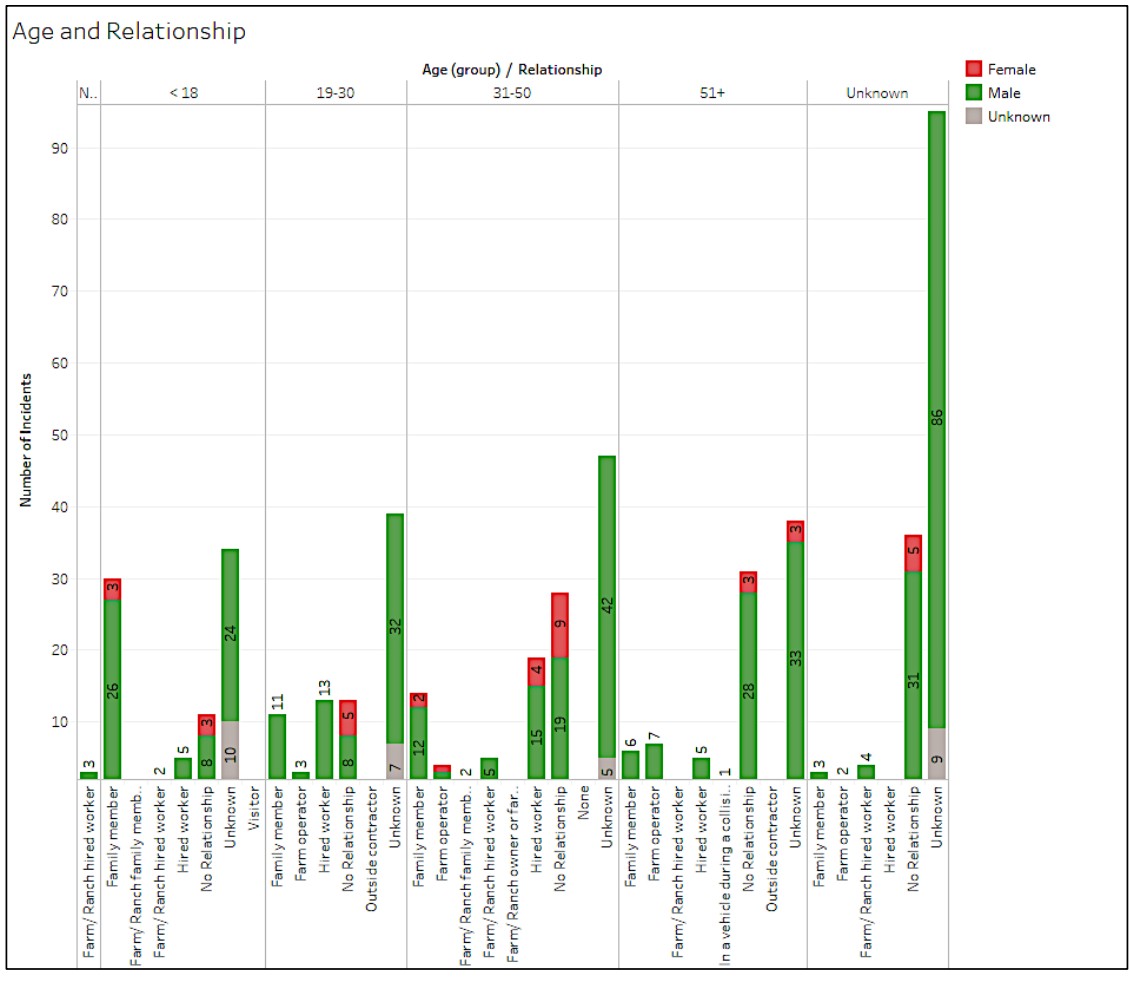

**Figure 9.** Dashboard software display of victim age and relationship to farm.

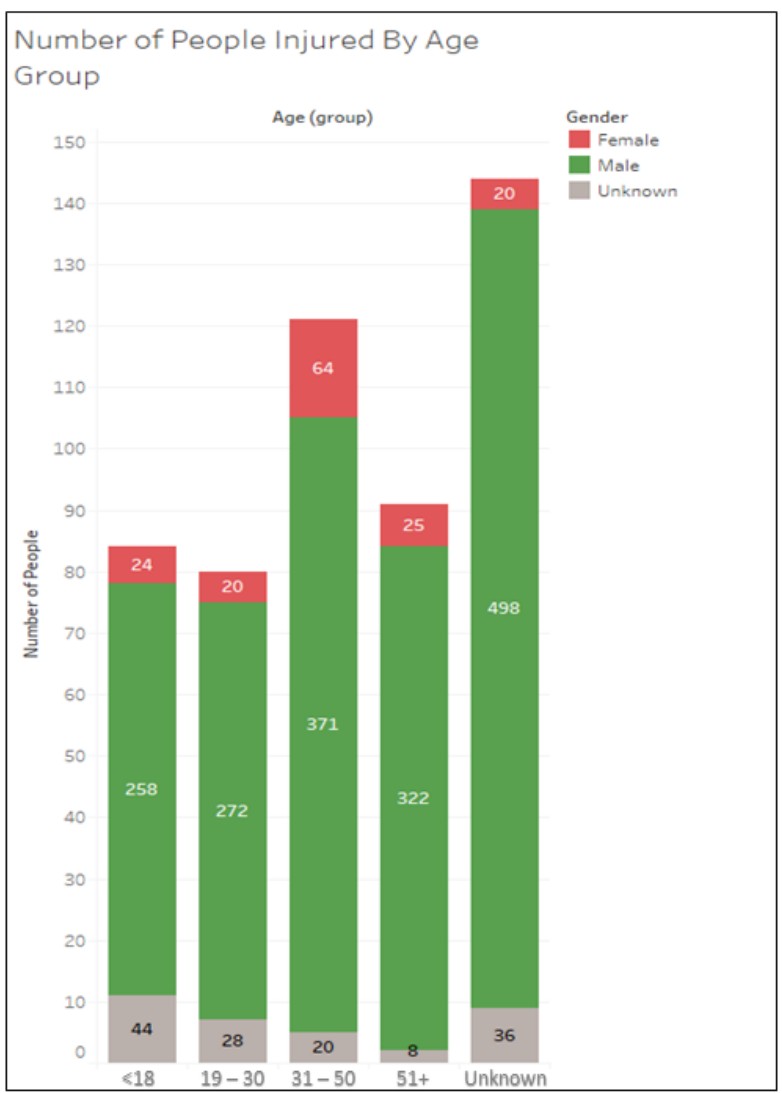

**Figure 10.** Dashboard software display of injuries and fatalities by gender.

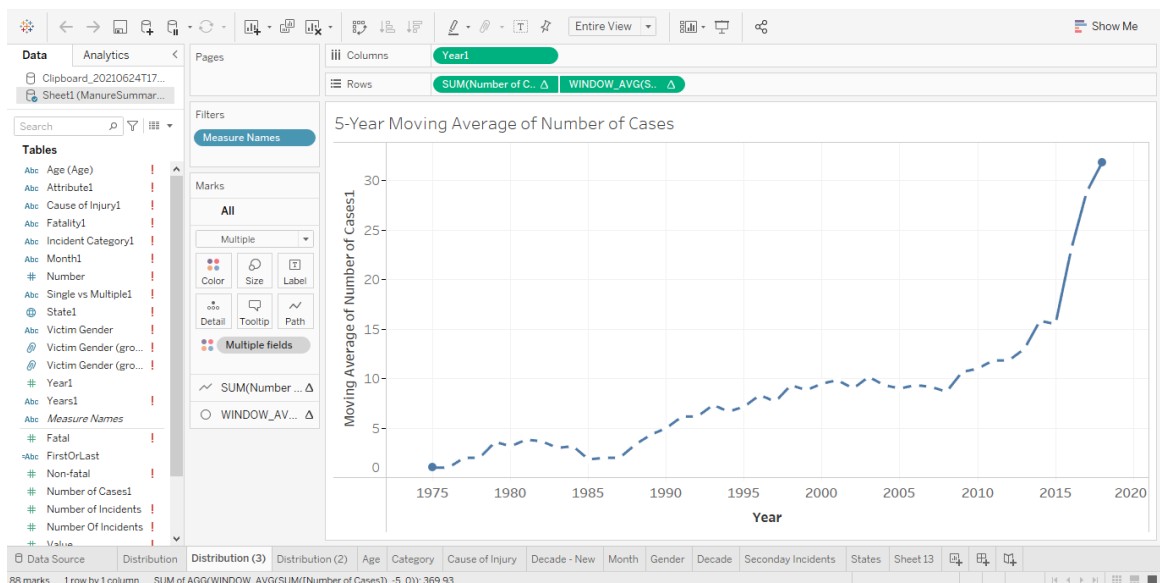

**Figure 11.** Dashboard software display of agricultural waste injury and fatality data, 1975–2020.

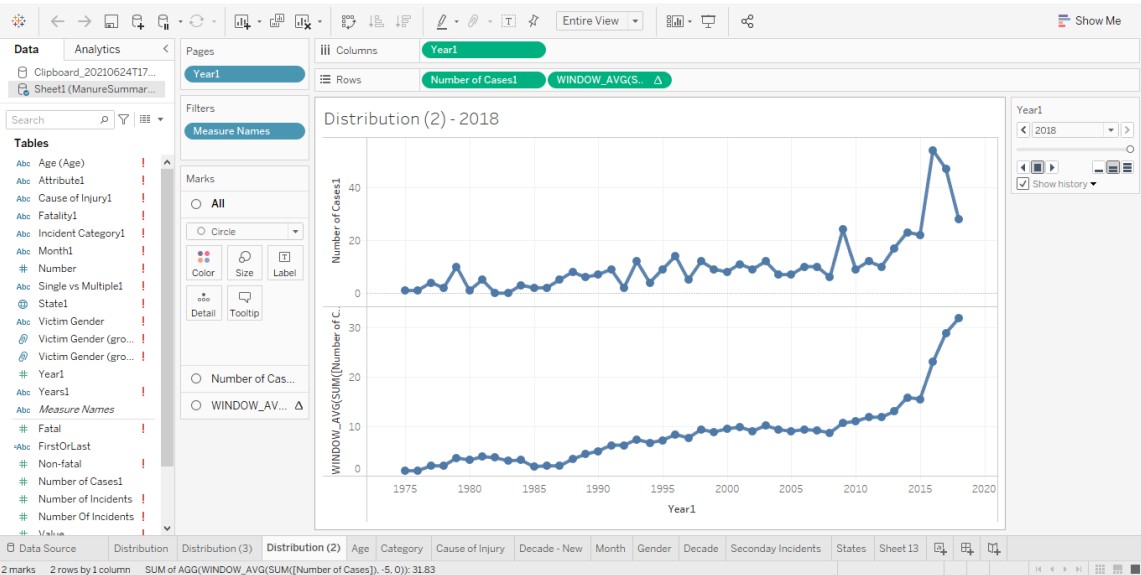

**Figure 12.** Dashboard software display of agricultural waste injury and fatality data.

## 5. Discussion and Evaluation

*Automated Data Collection and Record-Keeping for Monitoring*

The preliminary data used for this research incorporated 491 documented agricultural waste-related cases that were manually recorded by an experienced coder, and the coding process was based on the protocol developed by Nour [14]. This procedure was found to allow for classification of agricultural occupational injury and fatality data in real-time in a consistent manner. The major features of the demonstrated AgISM model were consistency, simplicity, and speed of injury coding process.

After the system was developed, the cases were inputted, and the data were visualized. Figure 11 shows the five-year moving average number of cases. It was clear from the data that the number of documented cases has been on the rise in recent times. A contributing factor to this increase was likely due to the more aggressive data collection efforts.

Furthermore, Figure 12 shows additional graphs that help visualize the cases that were documented over time.

Overall, the AgISM Model was demonstrated as a feasible monitoring tool that can help users visualize injury surveillance data to draw conclusions from past data and make future predictions in regards to cases and trends.

## 6. Conclusions

This research provided guidance for monitoring work-related injuries and fatalities that aim to build or improve automated surveillance of occupational-related incidents. The AgISM model provided data in near real-time for identifying trends for occupational safety and health programs to interact with other communities of interest. This article examined the application of the AgISM model to 491 documented cases that involved agricultural waste-related incidents for designing a novel automated injury and fatality case monitoring tool. Overall, the model performed as expected and could be implemented on similar datasets. In future work, as the size and theme of the dataset are changed, many factors such as the method of entering data, processing time, and data pre-processing can also be fully considered for ensuring the validity of the coding process.

With appropriate training for new coders, the coding process will be more reliable, and comparable, time-wise, with manual coding. This computer-environment coding tool for data from primarily news clipping sources can assist researchers or public health practitioners in preparing more accurate workplace injury reports in standardized fashion.

While it does take significantly more time to set up than manually coding data systems, in the long run, it prevents much of the need for data clean-up and extraction that has been historically required. It also allows for more rapid incorporation of new data as it becomes available.

Future work will further develop the AgISM model by enabling case record-keeping in real-time. Automation will be further integrated by relying on NLP (Natural Language Processing) in order to possibly allow automatic mining of information online and inputting into the database with minimal manual work. This can lead to a decrease in the number of unnoticed incidents and allow for further insights from the news clipping narratives.

Additionally, further work should be conducted to combine information from the database with regional variations, such as weather or crop yields, and additional information that can be deduced from the date and location of selected incidents. Statistical learning models can be generated to predict increases in the number of cases and allow for preventative measures based on new data. In this scenario, understanding the impact of climate change could allow for recommendations intended to offset the increased risk of work-related incidents.

Furthermore, data mining techniques could be applied to predict or gain insight into factors that could contribute to a potential increase in the frequency of incidents. This would require an analysis of correlations between potentially related information and the usage of techniques such as decision trees, regression, and random forests to create and test hypotheses. Understanding such factors in addition to automated monitoring categorizations can guide and support policy and decision makers with practical information to reinforce their decisions. It was noted that there is a good potential of successfully applying this model to nearly 3000 agricultural confined space incident reports currently categorized in the Purdue Agricultural Confined Space Incident Database (PACSID), and future documented incidents.

**Author Contributions:** Conceptualization, M.M.N.; methodology, M.M.N. and Y.M.A.; software, Y.M.A.; validation, M.M.N. and W.E.F.; formal analysis, M.M.N. and Y.M.A.; investigation, M.M.N.; resources, M.M.N., Y.M.A. and W.E.F.; data curation, M.M.N. and Y.M.A.; writing—Original draft preparation, M.M.N.; writing—Review and editing, M.M.N. and W.E.F.; visualization, Y.M.A.; supervision, W.E.F. and M.M.N.; project administration, M.M.N.; funding acquisition, M.M.N. and W.E.F. All authors have read and agreed to the published version of the manuscript.

**Funding:** This study was made possible by funding from Purdue's Agricultural Safety and Health Program (West Lafayette, Indiana) and the Central States Center for Agricultural Safety and Health (CS-CASH) (Omaha, Nebraska).

**Acknowledgments:** The authors would like to thank Ellen Duysen, (CS-CASH), UNMC College of Public Health who provided insight and expertise that greatly assisted this research. We are also grateful to Aaron Etienne, Ph.D. Candidate in Agricultural and Biological Engineering, Purdue University, for reviewing and editing the manuscript. Finally, the authors also thank Noah Haslett for his ongoing contributions in maintaining the PACSID database.

**Conflicts of Interest:** The authors declare no conflict of interest.

### Nomenclature

| | |
|---|---|
| **Case** | A documented incident involving one victim. |
| **Incident** | An injury, fatality, or near miss such as drowning, falling, or suffocation event that related to agricultural wastes. It could include one or more victims. |
| **Incident summary** | The narration used to record what happened through the event. |
| **Cause of injury** | Indicates the specific reason for an injury such as suffocation, drowning, trauma from falling, or entanglement in machinery. |

| Coding and classification system method | Is the most widely-used system for classifying the nature and external causes of injury. It means transforming the incident information gathered from different sources into useable form for surveillance purposes [21]. |
|---|---|
| Coding of data | Refers to the process of transforming collected information or observations to a set of meaningful, cohesive categories. It is a process of summarizing and representing data in order to provide a systematic account of the recorded or observed phenomenon [22]. |
| Coding scheme tool | A reliable tool to categorize detailed incident reviews that can be processed on the dataset. |

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
