# Peer review of "AgISM: A Novel Automated Tool for Monitoring Trends of Agricultural Waste Storage and Handling-Related Injuries and Fatalities Data in Real-Time"

_safety_

Round 1
Reviewer 1 Report
This manuscript reports on a study of fatalities and injuries from agricultural waste storage, handling, and transport operations. The manuscript is well-written, scientifically informative, and properly articulated. The authors are to be congratulated for their efforts. This topic is interesting, and related studies should be further explored. Below are some points for consideration for the authors.
1. Abstract, page 3: The authors used "outbreaks" to describe the occurrence of injury incidents. Please explain how to define an outbreak in these agricultural injuries. Detailed relevant explanations should thus be provided in the text.
2. Page 5, Line 198: More clarification about “three primary objectives” should be provided.
3. Page 5, Line 222: The author mentioned that the works of livestock waste storage, handling, or transport affect respiratory and machinery hazards in humans. Are there any other hazards related to these operations besides these two health hazards? If so, please specify.
4. Page 13, Line 388: In current practice, how long does it take from the occurrence of an occupational injury to the identification of the injury or injury trend by the relevant authority? Does the proposed model (AgISM model) only achieve near real-time in identifying trends? Is it possible to be real-time?
5. Page 14, Lines 404-408: The authors mentioned that the optimization of the original model could automatically obtain information on the occurrence of injuries. Automatically mining information online and input is important because manual typing may have problems with timeliness, accuracy, and other biases. However, the authors did not elaborate on the implementation of the proposed model. A detailed description of the development and application of this novel model should thus be provided.
Author Response
Responses to reviewer (1) comments
Reviewer 1
Comments and Suggestions for Authors
This manuscript reports on a study of fatalities and injuries from agricultural waste storage, handling, and transport operations. The manuscript is well-written, scientifically informative, and properly articulated. The authors are to be congratulated for their efforts. This topic is interesting, and related studies should be further explored. Below are some points for consideration for the authors.
- Response: Authors appreciated the comment that the topic and the efforts were, in general, well compiled and written. We agreed and support the comments suggesting that the related studies should be further explored along with the other valuable comments.
My specific comments are:
- Abstract, page 3: The authors used "outbreaks" to describe the occurrence of injury incidents. Please explain how to define an outbreak in these agricultural injuries. Detailed relevant explanations should thus be provided in the text.
- Response: In abstract: authors agree with the comment that “outbreaks” is not the proper term for describing the occurrence of injury incidents. Term “outbreaks” has been replaced to “patterns”.
- Page 5, Line 198: More clarification about “three primary objectives” should be provided.
- Response: As per this feedback, the authors have updated the Introduction section by adding additional information for the four primary objectives to the second last paragraph in a list format.
- Page 5, Line 222: The author mentioned that the works of livestock waste storage, handling, or transport affect respiratory and machinery hazards in humans. Are there any other hazards related to these operations besides these two health hazards? If so, please specify.
- Response: Yes, there are several other hazards related to agricultural waste operations such as drowning, exposure to pathogens, and fires and explosions due to pressure of methane gas. The other hazards related to livestock waste operations are added to the paragraph.
- Page 13, Line 388: In current practice, how long does it take from the occurrence of an occupational injury to the identification of the injury or injury trend by the relevant authority? Does the proposed model (AgISM model) only achieve near real-time in identifying trends? Is it possible to be real-time?
- Response: This is great feedback for the study. The focus of the current study was to develop a working model that will help provide a tool for documenting injury cases in agricultural occupations. A future study will further explore the time it takes for documenting and obtaining the cases for the possibility of providing a real-time system. As per this feedback, information regarding future works was added to the updated manuscript within the “Discussion and Evalutation” section and the updated “Conclusions” section.
- Page 14, Lines 404-408: The authors mentioned that the optimization of the original model could automatically obtain information on the occurrence of injuries. Automatically mining information online and input is important because manual typing may have problems with timeliness, accuracy, and other biases. However, the authors did not elaborate on the implementation of the proposed model. A detailed description of the development and application of this novel model should thus be provided.
- Response: In current situations, it usually takes over a year for official injury reports to be processed and reported and in some cases, longer. This comment relates to the issue authors raised earlier that there is no automatic inclusion of data to be classified and coded in real-time without a text reading method by the professional coder. Outdated data make it difficult of develop timely responses.

Reviewer 2 Report
Dear Editor
The study done by the authors is innovative and the overall article is well written, but there are ambiguities that should be resolved in the article:
Abstract
1. In the abstract of the article, you have explained a lot about the importance and purpose of the study. Therefore, it is better to summarize.
2. There is not much explanation about the method? What are the tools for this automatic and real-time system?
Introduction
3. Some sentences in the introduction need to be transferred to the method, Such as:
In this study, data from 491 agricultural waste-related injuries and fatalities were mined from the Purdue Agricultural Confined Space Incident Database (PACSID).
Related Works
4. In the "Related Works" section, which reviews similar studies; what measures have been taken to address the shortcomings of these studies in the present study? Such as: They are only capable of collecting and recording data with inaccuracies in the range of 72% - 88%.
5. What measures have you taken to reduce uncertainties such as: substantial variation, experience level, background, and level of comprehensiveness of the incident narrative or news clippings?
6. A general summary is necessary at the end of this section.
Materials and Methods
It is almost good
Framework of the AgISM Model
7. Explain more about "Boolean logic model"?
8. The focus of the discussion should be on the advantages and disadvantages of the study and the limitations of the study should be mentioned.
9. English language and style are fine/minor spell check required.
Author Response
Responses to reviewer (2) comments
Reviewer 2
Comments and Suggestions for Authors
Dear Editor
The study done by the authors is innovative and the overall article is well written, but there are ambiguities that should be resolved in the article:
Abstract
- In the abstract of the article, you have explained a lot about the importance and purpose of the study. Therefore, it is better to summarize.
- The authors agree with that comment and the abstract was revised and shortened.
- There is not much explanation about the method? What are the tools for this automatic and real-time system?
- Response: The information for the tools was added in the updated abstract as per the feedback from the reviewer.
Introduction
- Some sentences in the introduction need to be transferred to the method, Such as:
In this study, data from 491 agricultural waste-related injuries and fatalities were mined from the Purdue Agricultural Confined Space Incident Database (PACSID). (MN)
- Response: Authors agreed with the reviewers and the sentence in the introduction was transferred to the method section as per the feedback from the reviewer.
Related Works
- In the "Related Works" section, which reviews similar studies; what measures have been taken to address the shortcomings of these studies in the present study?Such as: They are only capable of collecting and recording data with inaccuracies in the range of 72% - 88%.
- Response: Regarding the inaccuracies of the current methods of data collection, the proposed process or model focuses on the summarization and near-real time displays of the data. The need to have an experienced trained coder is not changed. The diversity of coders and their lack of experience will continue to contribute to errors. Also, the problem of how accurate the reporting source including news reports, still remains a problem. Both of these issues are discussed.
- What measures have you taken to reduce uncertainties such as: substantial variation, experience level, background, and level of comprehensiveness of the incident narrative or news clippings?
- Response: This is great feedback. For this study. regardless of how great our data processing is, incident documentation still relies on the accuracy of the news reporter in getting the incident narrations right.
- A general summary is necessary at the end of this section.
- Response: A general summary sentence was added at the end of the related-work section.
Materials and Methods
It is almost good
Framework of the AgISM Model
- Explain more about "Boolean logic model"?
- Response: The definition of the Boolean logic mode in the context was explained in detail in a couple places in the manuscript.
- The focus of the discussion should be on the advantages and disadvantages of the study and the limitations of the study should be mentioned.
- Response: The advantages, disadvantages, and remaining problems, as stated in the response to comment #4 above, are addressed in the text. The key advantage is the capacity to present almost real-time summaries of the data.
- English language and style are fine/minor spell check required.
- Response: This is great feedback regarding language and style, we checked both in the manuscript.

Reviewer 3 Report
The topic under study in this manuscript can be interesting for readers of this journal. However, the authors should address some major shortcomings to improve the quality of their work.
1- The theoretical contributions of this study have not been highlighted in the Abstract section.
2- The authors should support many of the statements mentioned in the Introduction section with valid references. I recommend use
3- The research contributions and necessity for developing the approach should be discussed well in the Introduction section.
4- The Literature review section can be improved by adding more newly published studies.
5- Why did you your proposed method as a robust one? How can you validate the outputs of the model?
6- The practical implications of this study should be discussed in a separate section before the Conclusion.
7- The development suggestions of this study are unclear.
Author Response
Responses to reviewer (3) comments
Reviewer 3
Comments and Suggestions for Authors
The topic under study in this manuscript can be interesting for readers of this journal. However, the authors should address some major shortcomings to improve the quality of their work.
1- The theoretical contributions of this study have not been highlighted in the Abstract section.
- Response: The abstract has been updated as per the feedback
2- The authors should support many of the statements mentioned in the Introduction section with valid references. I recommend use
- Response: Specific references are provided. No additional references were identified that were related and the reviewer did not identify any in the comment.
3- The research contributions and necessity for developing the approach should be discussed well in the Introduction section.
- Response: The need for the model was expanded on the introduction, especially the benefits of more real-time reporting and identifying incident patterns.
4- The Literature review section can be improved by adding more newly published studies.
- Response: literature review section has been updated as per the feedback.
5- Why did you your proposed method as a robust one? How can you validate the outputs of the model?
- Response: Regarding the validation of the outputs of the AgISM model, this should be the next step of the research by conducting in-depth investigations of specific cases to see how they actually align with reported data.
6- The practical implications of this study should be discussed in a separate section before the Conclusion. Needs discussion
- Response: The authors discussed and evaluated the results in the “Discussion and Evaluation” section.
7- The development suggestions of this study are unclear.
- Response: the manuscript was revised for clarification as per feedback and feedback from other reviewers.

Round 2
Reviewer 1 Report
All my questions are well addressed.
Author Response
Thank you for your time!
Reviewer 2 Report
Necessary corrections have been made. Therefore, this article is acceptable.
Author Response
Thank you for your time!
Reviewer 3 Report
I appreciate the authors' efforts to consider my comments. But my comments have not been considered carefully in this round of review.
1- The abstract section should highlight the features of this model compared to existing ones focusing on the theoretical contributions of this study.
2- The Literature review section is still weak. As I stated before, this section should be improved by adding more newly published studies to demonstrate the current research gap exactly.
I also recommend that the authors reduce their self-citations and focus on providing a comprehensive literature review.
3- The outputs of this study should be discussed clearly.
Author Response
Continued feedback from Reviewer 3 is appreciated, and an attempt was made to address each concern.
- The abstract section should highlight the features of this model compared to existing ones focusing on the theoretical contributions of this study.
Response: The abstract is a very limited place to deeply dive into the comparison between the model developed by the authors and others, mostly undocumented approaches to summarizing data related to agricultural injuries. The only other known effort to collect and summarize agricultural injury data is Murphy, et al., (2019) which does not provide a real-time summarization of media sourced data. Their work is addressed in the review of literature.
Regarding the suggestion to expand the abstract to include discussion of the “theoretical contributions” of this study, the authors believe this is addressed in the opening sentences in stating the need for more reliable means of summarizing data, that such strategies, especially related to agricultural injuries is lacking and that the proposed model does provide a “near real-time surveillance tool to enable researchers to visualize data as it is being collected”.
Adding more to the abstract comes at a cost of removing other content that the other two reviewers indicated was appropriate.
- The Literature review section is still weak. As I stated before, this section should be improved by adding more newly published studies to demonstrate the current research gap exactly.
I also recommend that the authors reduce their self-citations and focus on providing a comprehensive literature review.
Response: The authors agree that the review of relevant literature appears weak; and more current published studies are needed. The reality unless the authors have missed some of these articles, which the reviewer appears to suggest they did, however, is that little recent work has been done on the surveillance, coding, summarization and displaying of agricultural-related injury data. Based upon the tens of millions of dollars spent by NIOSH on the problem of agricultural injuries, very little has been done to design a more effective surveillance process until recently with AgInjuryNews reported on by Murphy (2019) and Weichelt (2021). Neither, however, currently provide the means of providing real-time summarization and display. The authors disagree with the reviewer 3’s position that the authors should reduce their self-citations and focus of providing a comprehensive literature review. Reviewer 3 provides no specific example of current articles that were over looked, which would be gladly included if identified.
The concern over “Self-Citations” is unfounded, and the authors would have been neglect in nor at least citing all current work on injuries and fatalities associated with agricultural work. Most of this work, with few identified exceptions are associated with one or more of the authors.
A review of older published work was done and several works are now cited
- The outputs of this study should be discussed clearly.
Response: A re-read of the article was conducted by all the authors to enhance clarity. The manuscript was updated on lines 507 to 514:
“The primary goal of this research was to precisely track trends in agricultural incidences by using an essential automated tool in order to properly interpret the data and guide future prevention efforts. The outcomes of this study were evaluated by successfully addressing the four objectives that were identified. (1) The issues that are directly related to the conventional approach of manual-coding incidents were assessed. (2) A new pipeline was tested and developed. (3) A method for obtaining data summaries was created. (4) Finally, a visualization dashboard was created.”
Round 3
Reviewer 3 Report
The quality of this manuscript has been improved.